# Development and Validation of a Well-Being Measure on Young Basketball Players: The NFAPW Scale

**DOI:** 10.3390/ijerph17217987

**Published:** 2020-10-30

**Authors:** Jorge Lorenzo Calvo, Jorge García-González, Monica Flo García, Daniel Mon-López

**Affiliations:** 1Facultad de Ciencias de la Actividad Física y del Deporte (INEF—Sports Department), Universidad Politécnica de Madrid, 28040 Madrid, Spain; 2Facultad de Ciencias de la Actividad Física y del Deporte (INEF), Universidad Politécnica de Madrid, 28040 Madrid, Spain; jorge.garcia@upm.es; 3Facultad de Educación, Universidad Autónoma de Madrid, 28049 Madrid, Spain; monica.flo@estudiante.uam.es

**Keywords:** coach, player, psychology, questionnaire

## Abstract

Basketball is a sport in which, beyond the physical and technical skills, the psychological aspects are a decisive factor and could negatively affect the well-being of the player. The present study analyzes how 11 items belonging to two stakeholders (coach and player) could negatively affect the well-being of the athlete. A sample of elite young basketball players (*n* = 121) consisting of 55 males and 66 females, ranging in age from 16 to 23 (*M* = 20.12 ± 1.71), completed the Negative Factors Affecting Players’ Well-being (NFAPW) Questionnaire. This questionnaire was designed ad hoc and demonstrated good psychometric properties that confirmed that it is a valid and reliable instrument to measure how those factors negatively affect their well-being. The results showed that females have a greater perception of the factors that negatively affect their well-being, especially those related to the actions of the coach. However, no differences were found regarding the experience. Although this research provides an initial tool for measuring the well-being of the player during competition, future studies are encouraged to provide guidance to the coaches and players in dealing with the psychological variables in a better way.

## 1. Introduction

It would be appropriate to define well-being as a state of equilibrium or balance that can be affected by life events or challenges [1]. In the sports field, the impact on well-being has usually focused on the physical manifestation, rather than its cognitive management [2], as there is comparatively less research on the mental health of athletes. However, the interest in this subject is growing. It is known that the mental demands placed on athletes are a unique aspect of a sporting career, and these may increase their susceptibility to certain mental health problems [3].

Coaches and athletes are two of the most important human stakeholders of the game and relations between them have been studied in depth by the scientific community over recent years. The importance of the coaching climate on the well-being of young athletes has been clearly demonstrated [4,5], and researchers have focused on how the quality of the coach–athlete relationship is linked to athletes’ well-being [6,7]. Furthermore, the importance of these relations is even greater when dealing with young athletes [8,9] or different genders [10].

Young players attach great importance to how the coach transmits information and having a coach who generates a positive motivational climate in the team [11], as opposed to one whose ego is a negative characteristic [12]. Furthermore, a coach with a healthy interpersonal style improves the well-being of younger players [13], and his/her autonomy improves their motivation, self-confidence, and satisfaction.

A high-quality coach–athlete relationship is a fundamental precursor to athletes’ optimal functioning and promotes a series of positive circumstances, such as greater enjoyment and compromise [14,15]. There is evidence in the literature that personal dialogue with the coach enhances the athlete´s skills [16]. However, this dyad presents areas of emotional isolation, disagreements, and incompatibility [17]. Differences between genders with respect to the perception of interpersonal relationships are considered to be a determining factor in the relationship between coaches and players [18]. A study conducted in the UK on females revealed the powerful and often overbearing role of the coach and gendered ideologies concerning women’s sporting abilities [19]. Negative perceptions of coaches were also shown by female athletes concerning long and poor speeches or unexpected approaches [20]. 

Not only do the perceptions and actions of coaches cause negative effects, but the players’ own feelings about their personal state could compromise their well-being. The perception of pain, lack of hydration [21], lack of energy, fatigue [22], or sleep loss [23] can reduce his/her comfort during the game and directly affect the player’s stress, attention, or activation, among other psychological variables. In addition, previous literature has shown that interventions in five psychological areas could influence women basketball players [24]. These five topics are: (1) motivation, one of the driving forces that underlie the effort and dedication of the athlete and can be fueled by a wide variety of factors [25]; (2) stress, one of the most frequent causes of the abandonment of competitive children’s sports [26]; (3) self-confidence, an aid or an obstacle for the athlete [27]; (4) activation, important for the athlete to detect the responses they produce and know that these may be different depending on the stimulus [28]; and (5) attention, allows for the selection between internal and external stimuli that demand further processing.

Although there are studies that focus on the coach and young players [29,30], there is a lack of those that emphasize the negative effects, specifically in basketball. There is also no study found that simultaneously evaluates both stakeholders (coach and athlete). Consequently, the aim of the study was to design and validate a questionnaire (Negative Factors Affecting Well-being Players (NFAPW)) to measure the well-being of basketball players and track the negative psychological effects from two perspectives (coach and athlete).

## 2. Materials and Methods 

### 2.1. Participants

A total of 121 Spanish basketball players (55 males and 66 females) were included in the study. Participants’ ages ranged from 16 to 23 (*M* = 20.12, *SD* = 1.71). Non-Spanish speakers were excluded from the study. Two inclusion criteria were used to ensure the homogeneity of the sample: (1) all players must be actively competing in the national first league during the study, and (2) athletes had to have a minimum of five years’ experience. Participants’ actual experience ranged between 5 and 14 years (*M* = 8.57, *SD* = 4.93). 

### 2.2. Materials (Design of the First Version of the Questionnaire)

First, an exhaustive search of the previous literature was performed by two sports specialists. Following the psychological variables reported by Buceta [24] and Sampaio et al. [31], and according to Arribas [32], an ad hoc survey was created for this study. The two sports specialists selected a total of 16 items related to different situations in a match that could negatively affect the well-being of the players. In the case of a discrepancy, a third researcher acted as referee. The items were related to two blocks or stakeholders: on the one hand, the player’s well-being associated with the coach [18] and on the other hand, the well-being associated with the player’s own feelings [23] (see Table 1).

Players filled in the questionnaire, showing their degree of agreement on a 1–4 Likert scale. Moreover, they indicated the effect direction of the item [33] in the five categories previously selected (motivation, stress, self-confidence, level of activation, and attention).

### 2.3. Fieldwork

The first version of the questionnaire was evaluated by a committee of ten experts. As selection criteria, the experts had to meet the following requirements: (1) qualification as national basketball coaches, (2) basketball university teachers, (3) Spanish nationality, and (4) experience as coaches in national competitions of at least 10 years.

Content validation was performed in two successive rounds. In the initial phase, the questionnaire was sent to the committee of ten experts and contained 16 items with a Likert-type scale, in which they had to mark their degree of agreement using a Likert scale of 4 points (1–4). To eliminate an item from the questionnaire, a minimum of 25% of the experts should indicate a negative degree of agreement. The second version of the questionnaire was sent to an external committee of two experts in psychology to make a new review. After completing the first version of the questionnaire, the validity and reliability of the questionnaire were analyzed. Following the suggestions of Gómez-García et al. [34] and Taherdoost [35] for the development of instrumental studies, the approach for this task consisted of four distinct stages: (1) content validity analysis, (2) comprehension validity analysis, (3) construct validity analysis, and (4) reliability analysis of the final instrument. 

Once the validation process was completed, the results were analyzed to determine the influence of gender and players’ experience on their well-being. 

### 2.4. Procedure

The technique used for content validity analysis, following the recommendations of similar investigations [36,37], was expert judgment. After the content validity and the comprehension validity of the first version was reduced from 16 to 11 items at the end. A direct link to the final version of the questionnaire in Google Forms was sent to all the teams by email. The questionnaire was available online over 14 days to ensure that all players are in the same phase of the league season. After that date, no other surveys could be registered in the database. All the players were advised to fill in the questionnaire when they were relaxed and focused to avoid distractions and homogenize the response conditions. Players had an unlimited amount of time to complete the survey. Furthermore, athletes did not have a specific time in which to answer the test. Participation in the study was voluntary and anonymous. Informed consent was signed by the participants, and this study was approved by the ethics committee of the Polytechnic University of Madrid. Once the period to complete the questionnaire was closed, data collection began using the Google Forms tool. To facilitate the understanding of the athletes, instructions were added at the beginning of the questionnaire. 

### 2.5. Data Analysis

#### 2.5.1. Comprehension Validity

Comprehension validity was assessed by evaluating the standard deviation (SD) skewness and kurtosis, deleting items with a standard deviation score lower than 1 and skewness and kurtosis coefficients outside the range (−1 to 1) [38]. The discrimination level of each item was also examined using item-total correlation statistics. Those items with corrected item-total correlation values >0.20 and for which the elimination of the item did not substantially increase the reliability expressed by Cronbach’s alpha were considered adequate [39].

#### 2.5.2. Exploratory Factor Analysis (EFA)

The Kaiser–Meyer–Olkin (KMO) test and Bartlett’s sphericity test were calculated. Multivariate analysis was performed to identify meaningful subscales of the NFAPW. Principal components factor analysis (PCFA) was conducted with Varimax rotation using the criterion, for factor extraction, of eigenvalue >1 [38].

#### 2.5.3. Confirmatory Factor Analysis (CFA)

CFA on data was conducted using SPSS Amos Version 26.0 (IBM Corp, Armonk, NY, USA). We utilized comparative fit index (CFI), root mean square error of approximation (RMSEA), and standardized root mean square residual (SRMR) to assess model fit. CFI is considered optimal with values above 0.95. SRMR and RMSEA indicates a good model fit when less than 0.08 and 0.06, respectively [40].

#### 2.5.4. Reliability

Cronbach´s alpha was used to determine the internal consistency of the final version of NFAPW, following the recommendation of using an alpha above 0.70 to evaluate the adequacy of a reliability coefficient [41].

#### 2.5.5. Comparisons by Gender and Experience

The data were described with the arithmetic mean (M) and the *SD*. The Kolmogorov–Smirnov test was used to check the normality with results of *p* > 0.05 in all variables. A *t*-test was performed for independent samples when means of two groups were checked. When statistically significant differences were found, the effect size was estimated using the Cohen’s d index (*d*) [42], establishing two cut-off points: medium effect (0.30) [43] and large effect (0.60) [44]. Effect difference in terms of percentage was calculated % = ((M1 − M2)/M1) * 100. All analyses, except those related to CFA, were performed with IBM SPSS Statistics software (Version 25.0. IBM Corp.). The level of significance was set at *p* < 0.050.

## 3. Results

### 3.1. Content Validity Analysis

After analyzing the results obtained in each of the phases, following the evaluations of the experts, we eliminated those items that had a score of less than 7. After this phase, the second version of the questionnaire was made up of 13 items. Items 14, 15, and 16 were eliminated from the questionnaire.

### 3.2. Comprehension Validity Analysis

The comprehension validity analysis was performed and 12 of the 13 items were considered acceptable, deleting only item 11 (see Table 2).

The discrimination level of each item was also examined using item-total correlation statistics. Our data showed acceptable corrected correlation and Cronbach´s alpha values for 11 of the 12 items, supporting the elimination of item 10. Consequently, the final scale consisted of 11 items (see Table 3). 

### 3.3. Exploratory Factor Analysis (EFA)

The overall Kaiser–Meyer–Olkin (KMO) measure was 0.823, and Bartlett’s Test of Sphericity showed statistically significant results, 634.50 (*df* = 55, *p* < 0.001). These results allowed us to perform a principal components factor analysis (PCFA).

The dimensional structure of the instrument revealed, in accordance with the rotated matrix extracted, the existence of two factors. These two factors explained 62.079% of the total variance: factor 1 (36.954%) and factor 2 (25.125%) and were named Coach and Player (see Table 4).

### 3.4. Confirmatory Factor Analysis (CFA)

The model fit (*n* = 121), measured as the chi-squared/degrees of freedom ratio (χ^2^/*df*), was 0.945, presented the following fit indicators: CFI = 1.000, RMSEA < 0.001, and SRMR = 0.0396. This model (Figure 1) provided a very good fit.

### 3.5. Reliability Analysis

The analysis of Cronbach´s alpha showed good internal consistency for the NFAPW (0.813). The items of the first factor (coach) and second factor (player) obtained an excellent (alpha = 0.904) and a good (alpha = 0.795) internal consistency, respectively.

### 3.6. Comparisons by Gender and Experience

Regarding gender, females reported larger scores than males in the NFAPW scale score *t*_(119)_ = 3.32; *d* = 0.61; *%* = 21.46; *p* = 0.001 and in the coach dimension *t*_(119)_ = 3.32; *d* = 0.70; *%* = 32.62; *p* < 0.001. No differences were found in the player dimension *p* = 0.421 (see Table 5).

Concerning experience, no differences were found between the less experienced and more experienced players in any dimension (coach and player) or in the NFAPW scale score *p* > 0.05. (see Table 6).

## 4. Discussion

The two objectives of the study were to design a valid and reliable instrument to assess the factors that negatively influence the well-being of basketball players and to analyze the players’ gender and experience influence on these factors. Regarding these objectives, our results confirm the psychometric properties of the NFAPW and showed a gender influence on the perception of the factors that negatively affect the well-being of the players, both in the coach dimension (*p* < 0.001), and in the total scale (*p* = 0.001), with large effects [44].

According to the previous literature, our content and comprehension validation obtained adequate values [38]. The EFA determined that the two-factor model explained 62.08% of the variance. In addition, the CFA results provided optimum values of CFI, RMSEA, and SRMR that confirmed the excellent fit of the proposed model [40]. Moreover, the Cronbach’s alpha values obtained from each dimension, and from the total scale, ratified its internal consistency [41].

Hence, the NFAPW instrument allows for the evaluation of the player’s perception of the factors that negatively affect their well-being. This two-factor model determines which negative effects originate in the actions carried out by the coach or the players’ feelings. Furthermore, the NFAPW could help to determine the well-being of the players and facilitate the diagnosis of intervention programs with greater control.

The use of the NFAPW Questionnaire indicated well-being differences based on gender. Similarly, other studies suggest females are more vulnerable on a psychological level to external factors than men, especially when they feel that the complexity of the task increases [45], identifying the coach as a potential stressful external agent. Female players experience many stressful stimuli during the competition [46], the coach being one of them [47] and probably the most influential since the coach–athlete relationship is mutually stressful [44]. Variables like autonomy could be affected, and the outcomes would confirm the results of a study in which boys had higher levels of autonomy than girls, due to competence emerging as a predictor of flow [48]. Low self-confidence may lead the athletes to have insecurities and doubts in their decisions because they do not realistically know whether they are capable of coping effectively. This can cause the athlete to see the task as being too complicated with respect to the resources he or she has, generating a poor performance based on a poor perception of self-efficiency and not because of a lack of real skills [49].

In summary, females are more affected by the coach than males during competition. Thus, it would be very interesting to work on these variables, particularly when the player feels that they are not performing at the level that the activity demands, since their level of anxiety can be increased, generating imbalances that could decrease performance until there is an abandonment of the sport [50,51,52]. However, in the player dimension, the NFAPW did not show any significant differences between gender, probably because during the game, feelings of lack of sleep, tiredness, lack of hydration, energy, and muscle pain are common to both genders of basketball players.

Regarding the experience, our participants’ ages ranged from 16 to 23 (*M* = 20.12 ± 1.71) with a minimum of five years practicing basketball. We did not find significant differences between the less experienced and more experienced players in any dimension (coach and player). However, young athletes have a high level of intrinsic motivation so they may be more sensitive to the coach’s actions. According to the previous literature, early experiences influence the motivation to continue training in the long term [30]. Hence, the coach should know how to maintain the intrinsic motivation of each player to improve the training process and avoid these results.

It is important to highlight that the athletes are also stressors for the coach. Coaches should manage young athletes’ psychological needs [44]. It is known that coaches that have autocratic behaviors and are nervous or anxious can produce fear, hesitation, and doubt in players [53,54]. Conversely, coaches with self-awareness can have an amazingly positive impact on players [55]. Furthermore, a good motivational climate, avoiding negative feedback and supporting the autonomy of the player, is positively related to their basic psychological needs of competence, autonomy, and relatedness [56]. Accordingly, when the coach acts as a mentor during the training process, he/she can provide the necessary guidance for the younger players to reach higher levels [57]. In this line, athletes who experienced intense coach-related acute stress are more likely to have and need coping styles for stress [58]. Therefore, as long as the coaches avoid actions that produce stress in their players, they can positively affect their athletes’ well-being.

From the player dimension, acceptance-based interventions gained empirical support for the treatment of a broad range of psychological difficulties with athletes [59]. Internal cognitive and emotional states do not need to be eliminated to facilitate positive behavioral outcomes. Rather, it has been suggested that alternative strategies to improve the well-being may aim towards the acceptance of the present moment and internal experiences during the game [60], such as thoughts; feelings; physical sensations, like pain, fatigue, or lack of hydration or energy; enhanced attention to external responses; and contingencies that are required during the competition.

In summary, if the players intend to practice the sport in optimal conditions, psychological needs must be met, and there should be a self-determined motivation for practice, a positive relationship with the coach, knowledge at a professional level [8], family support, and the ability to effectively manage their own feelings as well as academics [12]. 

Even though the NFAPW questionnaire is valid and reliable for analyzing two stakeholders (coach and player), some limitations should be considered. These two aspects represent only part of the factors that can influence the well-being of basketball players. Moreover, an analysis of more sports levels and other age categories could improve our results. Accordingly, further studies with robust methodologies, including the analysis of different parameters that could affect the psychological well-being of the player, such as the referee, rivals, teammates, the public, or competition factors, are recommended by the authors of this study. As practical applications, advisory programs for coaches and players with the aim of improving the management of potentially stressful situations in competitions should be included. In addition, the use of questionnaires like the NFAPW could help coaches know their athletes better and individually influence their performance. Lastly, to achieve players’ optimal well-being, the NFAPW could be used by coaches to improve their knowledge of emotional intelligence and to sum up coping skills to take proper care of the player.

## 5. Conclusions

The NFAPW has proven to have good psychometric properties for assessing the negative factors affecting the well-being of young basketball players. This tool may be used to identify the actions taken by the players themselves or by the coach that influence the performance of the players.

The gender of the players influences their well-being. Female players have a greater perception of the factors that negatively affect their well-being, especially those related to the actions of the coach. However, no differences were found regarding experience.

## Figures and Tables

**Figure 1 ijerph-17-07987-f001:**
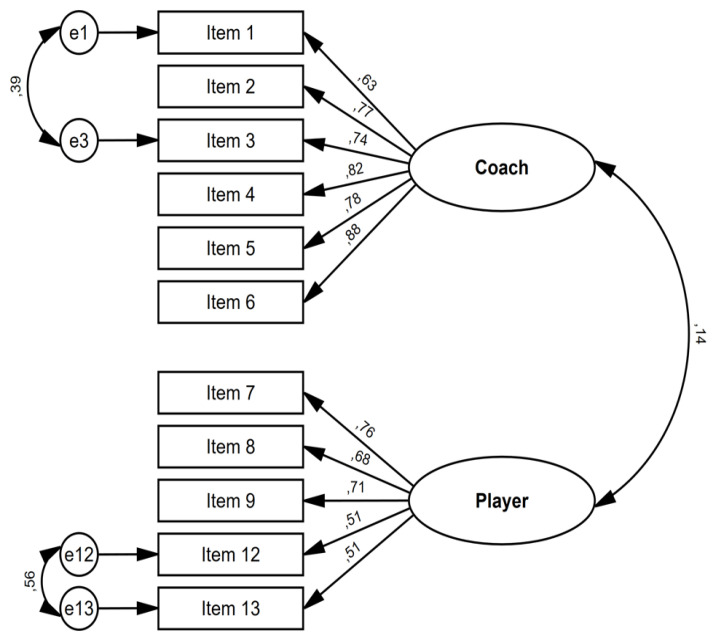
Confirmatory factor analysis for the Negative Factors Affecting Players’ Well-being (NFAPW).

**Table 1 ijerph-17-07987-t001:** Items of the first version of the questionnaire.

Item	Description
Item 1	Constant instruction from the coach during the match
Item 2	Not being in the team’s starting lineup
Item 3	Having the coach yell at you, telling you what you have done wrong
Item 4	Coach’s hasty decisions due to lack of experience
Item 5	That the coach overcomes a situation, during or after the game, due to pressure
Item 6	A lack of communication with the coach
Item 7	A feeling of exhaustion
Item 8	A lack of rest
Item 9	A sensation of physical pain
Item 10	The presence of family or friends in the stands
Item 11	The presence of a lot of ambient noise
Item 12	A lack of hydration
Item 13	A lack of energy
Item 14	Fear of failure
Item 15	Getting lost (in space)
Item 16	No clear roles established

**Table 2 ijerph-17-07987-t002:** Standard Deviation, skewness, and kurtosis indicators.

Item	SD	Skewness	Skewness SE	Kurtosis	Kurtosis SE
Item 1	0.973	0.059	0.22	−0.968	0.437
Item 2	1.024	0.360	0.22	−0.844	0.437
Item 3	1.037	0.015	0.22	−1.006	0.437
Item 4	1.033	0.221	0.22	−0.798	0.437
Item 5	0.931	0.687	0.22	−0.061	0.437
Item 6	1.142	0.450	0.22	−0.171	0.437
Item 7	0.974	−0.288	0.22	−0.938	0.437
Item 8	1.022	−0.354	0.22	−1.010	0.437
Item 9	1.025	0.112	0.22	−0.761	0.437
Item 10	1.082	0.966	0.22	−0.451	0.437
Item 11	0.728	1.762	0.22	2.691	0.437
Item 12	0.925	0.358	0.22	−0.752	0.437
Item 13	0.964	0.426	0.22	−0.722	0.437

*Note. n* = 121; SE = Standard Error; SD = Standard Deviation.

**Table 3 ijerph-17-07987-t003:** Discrimination index of the scale.

Item	Scale Variance If Item Deleted	Corrected Item-Total Correlation	Cronbach’s Alpha If Item Deleted
Item 1	38.195	0.525	0.773
Item 2	37.589	0.543	0.770
Item 3	36.795	0.603	0.764
Item 4	37.087	0.580	0.767
Item 5	37.591	0.613	0.765
Item 6	35.936	0.600	0.763
Item 7	39.967	0.369	0.787
Item 8	40.816	0.275	0.796
Item 9	38.994	0.423	0.782
Item 10	42.706	0.111	0.813
Item 12	40.463	0.352	0.789
Item 13	40.716	0.310	0.793

*Note*. *n* = 121.

**Table 4 ijerph-17-07987-t004:** EFA results for the NFAPW.

NFAPW Item	Factor Loading	
	1	2
Factor 1: Coach		
Item 6	**0.87**	0.03
Item 4	**0.83**	0.09
Item 3	**0.83**	0.06
Item 2	**0.82**	−0.01
Item 5	**0.81**	0.09
Item 1	**0.75**	0.04
Factor 2: Player		
Item 7	0.09	**0.76**
Item 13	−0.01	**0.75**
Item 12	0.02	**0.75**
Item 9	0.12	**0.74**
Item 8	−0.01	**0.69**

*Note*. *n* = 121. The extraction method was principal axis factoring with a Varimax rotation. Factor loadings above 0.30 are in bold.

**Table 5 ijerph-17-07987-t005:** NFAPW, coach dimension, and player dimension scores according to gender.

	Females ^a^		Males ^b^				95% CI		*Cohen’s*
Factor	M	SD	M	SD	*t* _(119)_	*p*	LL	UL	*d*
Coach	10.42	4.19	7.02	5.40	3.82	0.000 ***	1.64	5.18	0.70
Player	8.21	4.03	7.62	4.02	0.81	0.421	−0.86	2.05	0.15
NFAPW	18.64	6.23	14.64	7.00	3.32	0.001 **	1.62	6.38	0.61

*Note. *^a^*n* = 66. ^b^
*n* = 55. M = mean; SD = standard deviation; *t* = *t* value and degrees of freedom; CI = confidence interval set at 95%; LL = lower limit; UL = upper limit; *d* = Cohen’s d; ** *p* < 0.01. *** *p* < 0.001.

**Table 6 ijerph-17-07987-t006:** NFAPW, coach dimension, and player dimension scores according to experience.

	Higher ^a^		Lower ^b^				95% CI		*Cohen’s*
Factor	M	SD	M	SD	*t* _(119)_	*p*	LL	UL	*d*
Coach	9.21	4.89	8.58	5.22	0.69	0.494	−1.19	2.46	0.13
Player	7.79	4.07	8.08	4.01	−0.39	0.695	−1.74	1.17	−0.07
NFAPW	17.00	6.70	16.66	7.06	0.27	0.785	−2.14	2.83	0.05

*Note.*^a^*n* = 57. ^b^
*n* = 64. M = mean; SD = standard deviation; *t* = *t* value and degrees of freedom; CI = confidence interval set at 95%; LL = lower limit; UL = upper limit; *d* = Cohen’s d.

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
