# Peer review of "Development and Validation of a Well-Being Measure on Young Basketball Players: The NFAPW Scale"

_ijerph, 2020, doi:10.3390/ijerph17217987_

Round 1
Reviewer 1 Report
Overall the manuscript entitled, " Influence of Gender and Experience on the Well-Being of Young Basketball Players: A Pilot Study presented well with scientific support literature.
In the conclusion section, the author should more space to explain its outcome. This would be helpful for further research.
Reviewer 2 Report
An important focus for investigation in Sport Psychology. The outcomes are most relevant to the development of coaching behaviours and interactions. There are previous findings in relation to gender differences, however the outcomes of research such as is described in the article do add greatly to our knowledge and provide benefit to all sports participants, regardless of gender.
To focus on:
- The aim of the paper is unclear. The title suggests that it is research focused on gender, yet the paper gives considerable focus to instrument development in terms of validity and reliability. This is not a criticism as it is important for the validity of the findings. However I do recommend that the title, aims and the final conclusion are reconsidered in terms of instrument development as well.
- Line 38. Would consider changing the term ‘dimensions’. Coaches and athletes are not necessarily ‘dimensions’. I realise they are in the instrument, but this does not read well in the introduction. Suggest that they could be ‘stakeholders’ within the interactions. Yes they will be a dimension in the methods. Try not to get confusing here.
- Line 54 would reconsider the word ‘conditioner’ in the following sentence and rework
Differences between genders in respect to the perception of interpersonal relationships are considered to be a conditioner on the relationship between coaches and players
- In the procedure the paper refers to a second version of the questionnaire (Table 1), yet no development into a second version is mentioned prior?
- Lines 103 and 104 will need further explanation if they are to remain in the paper. What is occurring here is unclear.
- The versions of the questionnaire are alluded to but just not clear in the procedure. Assume this is part of the elimination of items as part of the analysis. Just go through and clarify this. It is fine for the analysis, however is confusing in other sections.
- Line 258 – use of the word ‘he’. I would either be gender neutral or use ‘he/she’
- Lines 264 – 269
Suggest that the following sentence is far too long and the reference at the end is not well placed. Recommend reworking
Rather, it is suggested that an alternative approach to the enhancement of athletic well-being may be achieved through strategies and techniques that target the development of present-moment acceptance of internal experiences during the game, such as thoughts, feelings, and physical sensations, like pain, fatigue, or lack of hydration or energy, and enhanced attention to external responses, and contingencies that are required during the competition [60].
- Fully recommend checking the whole document through for the correct or incorrect placement of each reference. There are places where the reference is added to your own findings. You will need to read through carefully (particularly results, discussion and conclusion) and delineate this. State you own findings and then apply the references by way of explanation or as support for previous findings as the case may be.
Reviewer 3 Report
INTRODUCTION
The introduction provides the background information for readers to understand the problem, but the importance on athlete’s performance of the psychological construct proposed.
The objectives must be placed at the end of the introduction
METHODS
More information about the specific characteristic of the participants would be necessary.
Statistical treatment is correct and well described
RESULTS
All of the tables and figure include specific and well developed statistic.
DISCUSSION
The discussion is according to the data obtained
Explain limitations of the study and possible practical applications.
The conclusion must be concise, responding objective of the study
LITERATURE CITED
The literature cited is relevant to the study
Reviewer 4 Report
1. Introduction
Lines 64-76: Be more direct with the aim of the study. Explain better.
2. Materials and Methods
There should be a section that explains the research design.
Line 88: What criteria have been followed to select these specialists? Explain the main characteristics.
2.3 Fieldwork
How was the expert judgment carried out?
What aspects were taken into account for content validity and expert judgment?
Detail the validation process by experts.
3. Results
3.1 Content Validity Analysis
Lines 148-154: The first two paragraphs are part of the methodological section: Fieldwork.
